# Effect of Frictional Slipping on the Strength of Ribbon-Reinforced Composite

**DOI:** 10.3390/ma14174928

**Published:** 2021-08-30

**Authors:** Yosyf Piskozub, Heorhiy Sulym

**Affiliations:** 1Department of Applied Mathematics and Physics, Ukrainian Academy of Printing, Pidgolosko 19, 79020 L’viv, Ukraine; 2Department of Mechanics and Applied Computer Science, Faculty of Mechanical Engineering, Bialystok University of Technology, Wiejska 45c, 15351 Bialystok, Poland; h.sulym@pb.edu.pl

**Keywords:** ribbon-like reinforcement, composite, thin inclusion, bimaterial, nonperfect contact, friction, jump functions, 00A06, 30E20, 45F15, 74M15, 74M25

## Abstract

A numerical–analytical approach to the problem of determining the stress–strain state of bimaterial structures with interphase ribbon-like deformable inhomogeneities under combined force and dislocation loading has been proposed. The possibility of delamination along a part of the interface between the inclusion and the matrix, where sliding with dry friction occurs, is envisaged. A structurally modular method of jump functions is constructed to solve the problems arising when nonlinear geometrical or physical properties of a thin inclusion are taken into account. A complete system of equations is constructed to determine the unknowns of the problem. The condition for the appearance of slip zones at the inclusion–matrix interface is formulated. A convergent iterative algorithm for analytical and numerical determination of the friction-slip zones is developed. The influence of loading parameters and the friction coefficient on the development of these zones is investigated.

## 1. Introduction

The theory and practice of the design and use of progressive composite materials with flat reinforcement provides indisputable evidence that their tensile strength in the transversal direction, in the case of unidirectional ribbon reinforcement, is between 50 and 75% of the strength in the longitudinal direction; the use of fibers typically yields only between 2 and 15% [1,2,3,4,5,6,7]. Reviews and monographs [3,4,7,8] note the advantages of flat reinforcement, which improves manufacturability and the mechanical properties of the composite, increases the reinforcement factor and the resistance to leakage failure, and reduces the statistical variation of the designed properties. It is a testament to the big prospective application of composites with ribbon-like reinforcement. The use of external ribbon-like reinforcement in steel-reinforced concrete makes it possible to save between 15 and 45% of metal in comparison with reinforced concrete and pure metal structures. In addition, thin ribbon-like elements are a common phenomenon in micro- and nanostructures [8,9,10].

In composite mechanics, two key issues can be distinguished: the determination of (1) the average effective properties of the composite as a whole, and (2) the elastic or plastic deformation processes and the possible failure of composite constituents, including contact loss at the matrix-filler interface. In the process of the exploitation of composites, the phenomenon of cracking and delamination is frequent; i.e., reinforcing heterogeneities can be in both ideal and non-ideal contact with the basic material, including at the interface of the media. Consideration of friction in the study of contact phenomena is one of the most pressing problems for mechanical engineering and materials science in the analysis of phenomena and processes occurring in the moving elements of machines, at various technological operations [11,12,13,14,15,16,17,18,19,20]. Thus, friction may be accompanied by electrical, thermal, vibrational, and chemical processes that dampen internal dynamic processes, significantly affecting the intensity in the wear and tear of materials and, consequently, the reliability and durability of the structural elements made of them [12,13,16,19]. The influence of friction can be both negative and positive.

From the point of view of structural integrity mechanics, friction between crack faces at their relative displacement is useful in most cases, since it causes dissipation of internal strain energy; thus, it reduces stress concentration, and reduces or even eliminates alternating plastic deformation under an alternating load. It is also known that the development of the residual stress field contributes to the adaptation of the material to operational loads. The compression of composite materials arising under the action of frictional forces improves the redistribution of shear stresses, even in the case of macroscopic failure of the reinforcement-matrix interface.

The negative consequences of friction are mainly the wear and tear on the contact surfaces, as well as thermal radiation. At an excessive intensity, the latter can sometimes cause unpredictable changes in the mechanical, physical, and chemical properties of the material and the distribution of physical fields, influencing diffusion processes, in particular hydrogen diffusion, and the development of fracture phenomena.

Most progressive multiscale methods [4,9,10] recognize a priori that the overall behavior of a composite is highly dependent on the local details. For example, local imperfections in the spacing and direction of fibers or ribbons can negatively affect the overall bearing capacity of a structure consisting of a composite material.

However, it is impossible to draw reliable conclusions about this, especially to optimize the stress–strain state of the structure using direct numerical methods such as the finite element method. Therefore, the development of prediction methods that can adequately reflect the complex mechanical behavior near such inhomogeneities is a challenge and requires the application of either analytical tools or numerical–analytical methods.

The need to take into account the aforementioned geometric nonlinearity (a priori unknown contact spots) significantly complicates the process of problem-solving and requires the use of various approximate methods, even for bodies of simple geometry [4,6]. However, these methods do not always allow for the correct consideration of thin-walled heterogeneity, nor do they guarantee the accuracy of a solution for load optimization in future applications. During the operation of materials that have the structure that is being considered in this work, especially when loaded by concentrated force and dislocation factors, undesirable critical states of adhesion loss between the constituent parts may arise. This can be avoided by locating the loading points in the so-called “safety zone” when the shear stresses on the contact surfaces at each of their points do not exceed the critical value.

It should be noted that the vast majority of research studying the structural stability of bodies with thin inhomogeneities does not cover the entire possible range of force or dislocation loading of structures. Additionally, it does not provide a full opportunity to determine the critical load and so-called “safety zone” for applications in order to optimize the properties of structures for certain types of loading.

This work aims to develop a numerical–analytical method to study structures with ribbon-like deformable elements, with possible frictional contact between the constituent elements; moreover, it aims to study the mechanical effects of loading by force and dislocation factors on its strength.

## 2. Formulation of the Problem

Consider an unbounded isotropic bulk consisting of two half-spaces with the elastic constants $E_{k},\;\nu_{k},\;G_{k}\;(k=1,2)$, pressed to the interface by normal stresses $\sigma_{yy}^{\infty }<0$, and under the action of uniformly distributed at infinity stresses $\sigma_{xxk}^{\infty }\;(k=1,2)$. The external longitudinal shear load is determined by the stresses $\sigma_{yz}^{\infty }$ and $\sigma_{xzk}^{\infty }$ uniformly distributed at infinity, concentrated intensity forces $Q_{k}$, and screw dislocations with the Burger’s vector component $b_{k}$ at points $\varsigma_{*k}\in S_{k}$ (k=1, 2), oriented along the axis in such a way that their action causes a quasi-static antiplane SSS in the body. To ensure the straightness of the material interface at infinity, the stresses must satisfy the conditions $\sigma_{xz2}^{\infty }G_{1}=\sigma_{xz1}^{\infty }G_{2}$, $\frac{\nu_{2}\sigma_{yy}^{\infty }-\left(1-\nu_{2}\right)\sigma_{xx2}^{\infty }}{G_{2}}=\frac{\nu_{1}\sigma_{yy}^{\infty }-\left(1-\nu_{1}\right)\sigma_{xx1}^{\infty }}{G_{1}}$.

We will study the SSS of the body section with a plane xOy perpendicular to the direction Oz of its longitudinal displacement. The plane sections of half-spaces perpendicular to this axis form two half-planes $S_{k}\;(k=1,2)$, and the abscissa axis corresponds to the interface L~x between them (Figure 1).

Along the segment $L^{\prime }=[-a;\;a]$ is a thin inclusion of thickness 2h (h≪a) (Figure 1), of which the upper and lower banks may come in contact with the matrix non-ideally on the intervals ${L^{\prime \prime }}^{\pm }=[b^{\pm };\;c^{\pm }]\;\left(\left|b^{\pm }\right|\leq \left|a\right|,\left|c^{\pm }\right|\leq \left|a\right|\right)$, respectively. $G_{x}^{in}$ and $G_{y}^{in}$ are the shear moduli of the inclusion material. The upper index “in” denotes the values describing the inclusion material’s SSS.

The contact between the half-spaces along the line $L\backslash L^{\prime }$ and at the inclusion–matrix interface at the sections $L^{\prime }\backslash L^{\prime \prime }$ is also mechanically ideal:(1)$$w(x,+0)=w(x,-0),\;\sigma_{yz2}^{}(x,+0)=\sigma_{yz1}(x,-0),\;x\in L\backslash L^{\prime }$$
(2)$$\begin{array}{l} w(x,-h)=w^{in}(x,-h),\;\sigma_{yz}^{in}(x,-h)=\sigma_{yz1}(x,-h),\;x\in L^{\prime }\backslash {L^{\prime \prime }}^{-},\\ w(x,h)=w^{in}(x,h),\;\sigma_{yz}^{in}(x,h)=\sigma_{yz2}(x,h),\;x\in L^{\prime }\backslash {L^{\prime \prime }}^{+}. \end{array}$$

At the contact areas ${L^{\prime \prime }}^{\pm }$, we assume stick-slip contact conditions [18], wherein mutual slippage of contacting body surfaces can start, causing heat release, energy dissipation, wear [11,16,17,18], etc., and that all points of ${L^{\prime \prime }}^{\pm }$, tangential stresses (friction forces) are equal to:(3)$$\sigma_{yz}^{in}(x,\pm h)=\sigma_{yz2}^{}(x,\pm h)=-\mathrm{sgn}\left(w^{in}(x,\pm h)-w(x,\pm h)\right)\tau_{yz}^{\max}(x),\;$$
where $\tau_{yz}^{\max}(x)=-\alpha \sigma_{yy}\;(x)\;\left(\sigma_{yy}<0\right),$
α is the sliding friction coefficient. Outside the area ${L^{\prime \prime }}^{\pm }$, in the absence of slip on the inclusion surface, the tangential stresses may not exceed the allowable maximum
(4)$$\left|\sigma_{yz}(x,\pm h)\right|\leq \tau_{yz}^{\max}(x)\;\left(\sigma_{yy}<0\right)$$
and there is no mutual displacement of contact surfaces (displacement jump). The sign (direction of action) of the tangential stresses is chosen depending on the sign of the displacement difference $w^{in}(x,\pm h)-w(x,\pm h)$ at the point in question at $L^{\prime \prime }$.

In the case of normal pressure in the first approximation, we obtain:(5)$$\tau_{yz}^{\max}(x)=-\alpha \sigma_{yy}^{\infty }$$

It is not difficult to obtain a more exact expression by constructing the solution of the corresponding plane problem. The application of friction law in classical form (5) gives an opportunity, of course, to simplify boundary conditions for the main problem, but the choice of complex friction models [11,12,13,14,17], considering wear, will not fundamentally complicate the solution process.

## 3. Materials and Methods

In general, the formulated problem contains three component modules: an internal problem (the stress–strain state in the inclusion), an external problem (the stress–strain state in the matrix), and the contact conditions (1)–(4), which relate them.

The stress–strain state in a thin inclusion (the internal problem) is described by an appropriate mathematical model. Due to the small thickness of the inclusion, it is possible to construct approximate relations between the components of the stress tensor and displacement vector on the opposite sides of the inclusion, which adequately describe its SSS. For example, a rather general model of a physically nonlinear thin inclusion is built in [6]:(6)$$\left\{ \begin{array}{l} G_{x}^{in}(\sigma_{xz}^{in}){\langle \frac{\partial w^{in}}{\partial x}\rangle }_{h}(x)-2\sigma_{xz}^{in}(-a)-\;\frac{1}{h}\int_{-a}^{x}{\left[\sigma_{yz}^{in}\right]}_{h}(\xi )d\xi =0,\\ G_{y}^{in}(\sigma_{yz}^{in}){\left[w^{in}\right]}_{h}(x)+h{\langle \sigma_{yz}^{in}\rangle }_{h}(x)=0, \end{array}\right.$$
where $G_{x}^{in}$ and $G_{y}^{in}$ are the variable shear moduli of the inclusion material. Taking them to be constant, we obtain a special case of Hooke’s law. Hereinafter, the following notations are used:$\;{\left[\varphi \right]}_{h}=\varphi \left(x,-h\right)-\varphi \left(x,+h\right)$ and $\;{\langle \varphi \rangle }_{h}=\varphi \left(x,-h\right)+\varphi \left(x,+h\right)$; the indexes “+” and “−” correspond to the limit values of functions at the upper and lower edges of the line L. 

To solve the external problem, it is convenient to apply the method referenced in [6,7], which uses the well-known jump function method (JFM). According to its paradigm, a thin inclusion in the matrix is modeled by jumps of the components of the stresses and displacement vectors on the line ${L^{\prime }}_{}$:(7)$$\begin{array}{l} {\left[\sigma_{yz}\right]}_{h}≅\sigma_{yz}^{-}-\sigma_{yz}^{+}=f_{3}^{}\left(x\right),\;\\ {\left[\frac{\partial w}{\partial x}\right]}_{h}≅\frac{\partial w^{-}}{\partial x}-\frac{\partial w^{+}}{\partial x}={\left[\frac{\sigma_{xz}}{G}\right]}_{h}\equiv \frac{\sigma_{xz}^{-}}{G_{1}}-\frac{\sigma_{xz}^{+}}{G_{2}}=f_{6}^{}\left(x\right),\;x\in {L^{\prime }}_{}^{}; \end{array}$$
(8)$$f_{3}^{}\left(x\right)=f_{6}^{}\left(x\right)=0,\;\mathrm{if}\;x\notin L^{\prime }.\;$$

Furthermore, we can obtain the dependences according to which components of the stress tensor and the derivatives of the displacement vector inside an unbounded plane S obtain the form:(9)$$\begin{array}{l} \sigma_{yzk}\left(\varsigma \right)+i\sigma_{xzk}\left(\varsigma \right)=\sigma_{yzk}^{0}\left(\varsigma \right)+i\sigma_{xzk}^{0}\left(\varsigma \right)-\;\frac{G_{1}G_{2}}{G_{1}+G_{2}}\frac{1}{\pi }\int_{{L^{\prime }}_{}}\frac{f_{6}^{}\left(\xi \right)d\xi }{\xi -\varsigma }+\\ +i\frac{G_{k}}{G_{1}+G_{2}}\frac{1}{\pi }\int_{{L^{\prime }}_{}}\frac{f_{3}^{}\left(\xi \right)d\xi }{\xi -\varsigma }\;\left(\varsigma =x+iy,\;\varsigma \;\in S_{k};\;r=3,\;6;\;k=1,\;2\right), \end{array}$$
and their boundary values on the upper and lower banks of the line L are the following:(10)$$\begin{array}{l} \sigma_{yzk}^{\pm }\left(x,\pm h\right)=\mp \frac{G_{k}}{G_{1}+G_{2}}f_{3}^{}\left(x\right)\;-\frac{G_{1}G_{2}}{G_{1}+G_{2}}\frac{1}{\pi }\int_{{L^{\prime }}_{}}\frac{f_{6}^{}\left(\xi \right)d\xi }{\xi -x}+\sigma_{yz}^{0\pm }\left(x,\pm h\right),\;\\ \sigma_{xzk}^{\pm }\left(x,\pm h\right)=\mp \frac{G_{k}}{G_{1}+G_{2}}f_{6}^{}\left(x\right)+\frac{G_{1}G_{2}}{G_{1}+G_{2}}\frac{1}{\pi }\int_{{L^{\prime }}_{}}\frac{f_{3}^{}\left(\xi \right)d\xi }{\xi -x}+\sigma_{xz}^{0\pm }\left(x,\pm h\right).\; \end{array}$$

The values, further marked with the index “0” on top, correspond to the bulk’s SSS model without inhomogeneities (inclusions, cracks, etc.) under the corresponding external load (homogeneous solution). Hereinafter, the notations [6] are used:(11)$$\begin{array}{l} \sigma_{yzk}^{0}\left(\varsigma \right)+i\sigma_{xzk}^{0}\left(\varsigma \right)=\tau +i\{\tau_{k}+D_{k}\left(\varsigma \right)+\frac{G_{k}-G_{j}}{G_{1}+G_{2}}{\bar{D}}_{k}\left(\varsigma \right)+\frac{2G_{k}}{G_{1}+G_{2}}D_{j}\left(\varsigma \right)\},\;\\ D_{k}\left(\varsigma \right)=-\frac{Q_{k}+iG_{k}b_{k}}{2\pi \left(\varsigma -\varsigma_{*k}\right)}\;\left(\varsigma \in S_{k},\;k=1,\;2;\;j=3-k\right). \end{array}$$

Additional balance conditions must be imposed on the solution of the external problem:(12)$$\begin{array}{l} \int_{-a}^{a}f_{3}^{}\left(\xi \right)d\xi =\;2h\left(\sigma_{xz}^{in}(a)-\sigma_{xz}^{in}(-a)\right),\;\\ \int_{-a}^{a}f_{6}^{}\left(\xi \right)d\xi =\left[w\right](a)-\left[w\right](-a). \end{array}$$

Equation (10) can be used directly to determine the critical values of the load applied to the structure at which slippage will begin. At some point(s) ${L^{\prime \prime }}^{\pm }$ when the maximum allowable slippage $\tau_{yz}^{\max}(x)$ is reached:(13)$$\sigma_{yzk}^{}\left(x,\pm h\right)=\tau_{yz}^{\max}(x),\;$$
that is, once the expressions for the jumps $f_{r}^{}\;(r=3,6)$ are known, the applied load can be investigated and its critical values determined.

It is possible to apply the classical JFM to the solution of the obtained system of Equations (1)–(12), which provides the substitution of (10) into (6) using (1)–(4), and obtains the resulting system of singular integral equations (SSIE) to determine the unknown $f_{r}^{}\;(r=3,6)$ and the stresses in the matrix using (9) and inside the inclusion using (1)–(4). However, such a scheme for solving the problem works well in the case of an ideal contact of the structure components, simple geometry, and linear constitutive properties of the inclusion material. In the case of a non-ideal contact, with the a priori unknown dimensions of the slip zones $\left(c^{\pm }-b^{\pm }\right)$, rather complicated algorithms for solving the SSIEs are required, which do not always guarantee calculation accuracy. 

We propose a different approach to solving such a problem, which can be called a structurally modified JFM. The idea is to combine all equations into a global system without substituting the boundary conditions (1)–(4) into the model Equation (6), and limit values of matrix components of the stress–strain relations (9). Furthermore, it is convenient to solve this system of equations by any numerical–analytical method, for example, by the collocation method. Submitting the system of Equations (1)–(12) in discrete form in the set of collocation points $\;\left(x_{n},\;n=\bar{1,N}\right)\;$, we obtain the system of 6N linear algebraic equations (SLAE) for the determination of 6N unknowns $\sigma_{yz}^{in}(x_{n},\pm h),\;\frac{\partial w^{in}}{\partial x}(x_{n},\pm h),\;f_{r}^{}(x_{n})\;\left(r=3,6;\;n=\bar{1,N}\right)$. Of course, the number of unknowns for solving the problem increases, which is not crucial with modern computational capabilities. However, the construction of SLAEs is considerably simplified, as its modularity allows for making independent changes in separate modules with significantly less effort than constructing and solving new classes of problems with considerably more complicated parameters.

## 4. Numerical Results and Discussion

The accuracy of the solution in the case of non-ideal contact is very sensitive to the correct determination of the position and size of the slip zones. Within the framework of the proposed structural–modular MFS, we apply an iterative approach for this at each point of the interval ${L^{\prime \prime }}^{\pm }$: (1) gradually with growth, we apply a small load to check condition (13) of the beginning of the slippage process; (2) as soon as condition (13) is satisfied at certain points $x_{n}$, we assign values $\tau_{yz}^{\max}$ to values $\sigma_{yz}^{in}(x_{n},\pm h),\;\sigma_{yzk}^{}(x_{n},\pm h)$ in all these points and re-solve the SLAE; (3) we check whether constraint (4) is satisfied everywhere; if not, we assign values $\tau_{yz}^{\max}$ again to values $\sigma_{yz}^{in}(x_{n},\pm h),\;\sigma_{yzk}^{}(x_{n},\pm h)$ in those points $x_{n}$ where (4) is not satisfied. The process is repeated until condition (4) is satisfied in all points $\;\left(x_{n},\;n=\bar{1,N}\right)\;$. It is proven that such an iterative algorithm is convergent under monotonically increasing non-contrast loading. Despite the increased number of SLAE equations, the calculations showed that a relative error of 0.1% of the results can be achieved already at 21 collocation points for no more than seven iteration steps in the worst case of a very stiff inclusion. The obtained results were validated by comparing them with known partial solutions for an interfacial crack and an interfacial thin rigid inclusion, as well as a thin elastic interfacial inclusion at its ideal contact with the matrix [6,7,15,18].

Figure 2 and Figure 3 show the results of the study of the “safety zone” for the intensities and coordinates of concentrated forces $({\tilde{Q}}_{2}=-{\tilde{Q}}_{1}=\tilde{Q},\;$ $\;{\tilde{\varsigma }}_{*k}=\varsigma_{*k}/a={\tilde{x}}_{*k}+i{\tilde{y}}_{*k},\;$ ${\tilde{\varsigma }}_{*2}={\bar{\tilde{\varsigma }}}_{*1},\;$ ${\tilde{y}}_{*k}=\pm i\tilde{d})$, where $\tilde{Q}=Q/\pi aG_{av}$, ${\tilde{x}}_{*2}=x_{*2}/a,\;$ $\tilde{d}=d/a,\;$ $G_{av}=\left\{\sqrt{G_{1}G_{2}},\;\max\left(G_{1},\;G_{2}\right),\;Q/\pi a\right\}$. The “safety zone” for a concentrated factor of a certain value will be understood as the coordinates of its application at which slippage does not yet start at any point of the boundary ${L^{\prime }}_{}$; i.e., condition (4) is not fulfilled. We define the boundary of the “safety zone” from the following criterion: condition (4) starts to be fulfilled at least at one point. It can be argued that an inclusion harder than the matrix changes the form of the “safety zone” much less than a softer one. The particular case of no inclusion ${\tilde{G}}_{y}^{in}={\tilde{G}}_{k}\;\left({\tilde{G}}_{y}^{in}=G_{y}^{in}/G_{av};\;{\tilde{G}}_{k}=G_{k}/G_{av}\right)$ shows the coincidence of the results with those obtained in [6,7,18]. Additionally, note the expected trends of linear dependence of the growth of the critical value ${\tilde{Q}}^{*}$ on the increase in the value ${\tilde{\tau }}_{yz}^{\max\pm }=\frac{\tau_{yz}^{\max\pm }}{G_{av}}$, as well as on the increase in the distance $\tilde{d}$ of the points of application of the concentrated force. These effects are especially appreciable for the “soft” inclusion, when ${\tilde{G}}_{y}^{in}\ll {\tilde{G}}_{k}$.

Figure 4, Figure 5, Figure 6, Figure 7, Figure 8 and Figure 9 show the effect of slip on the stress ${\tilde{\sigma }}_{yz}^{}=\sigma_{yz}^{}/G_{av}$ distribution along with the inclusion–matrix interface, as well as the growth of the slip zone size and its intensity ${\tilde{w}}_{sl}=w_{sl}/a$ depending on the problem parameters. It is noteworthy that for a softer than the matrix inclusion, the slippage appears and grows faster than for a more rigid inclusion with the same problem parameters (Figure 4, Figure 6 and Figure 7). Decreasing the distance of the application points from the inclusion axis, as well as increasing the load intensity, is expected to increase the slip area and its magnitude (Figure 5, Figure 6 and Figure 7 and Figure 9). However, the displacement of the coordinates of the force application points along the inclusion axis to its apex (Figure 8), as well as the distance from this axis (Figure 5), reduces the slip intensity. The loading by an applied screw dislocation with a Burger’s vector ${\tilde{b}}_{2}=b_{2}G_{2}/G_{av}$ in p. $x_{*2}=0;\;y_{*2}=a$ leads to the appearance of two slip zones antisymmetric for the vertical axis of inclusion (Figure 9).

## 5. Conclusions

The proposed numerical–analytical approach to study the effects of mechanical contact imperfections and friction effects of a bimaterial structure with an interfacial thin ribbon-like deformable inclusion made it possible to obtain some important new results. Firstly, the a priori unknown configuration of the sliding friction zone was determined under different types of loads.

It is revealed that with the friction coefficient, type, and place of load application unchanged, the slippage will start at a lower load and the slippage zone will be larger in the case of a harder than the matrix inclusion. It can be argued that the shape and size of the “safety zone” changes much less in the case of a harder than matrix inclusion than in the case of a softer one.

The calculations confirmed the linear attenuation of the sizes of the safety and slip zones from a decrease in the intensity or distance from the inclusion of the concentrated factors. This is especially noticeable for the softer inclusion than the matrix inclusion.

The distribution of stresses in the inclusion vicinity, calculated to optimize the load regime of the investigated structure, makes it possible to conclude that the greatest influence on its stress–strain state is in the inclusion vicinity; accordingly, the appearance of slippage between the components has the case of applying concentrated forces in the point above the inclusion center.

The obtained results and the proposed method can be used and developed to study the value of the energy dissipated on the inclusion, heat release due to friction, development of wear processes, and the interaction of band inhomogeneities in composite structures with subsequent determination of their effective mechanical characteristics, optimization of operating load modes and the like.

## Figures and Tables

**Figure 1 materials-14-04928-f001:**
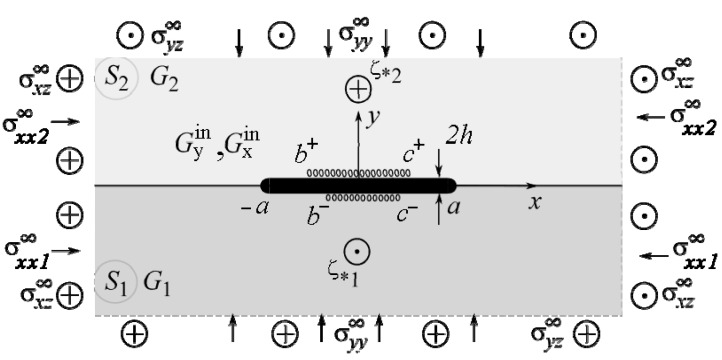
Geometry and load pattern of the problem.

**Figure 2 materials-14-04928-f002:**
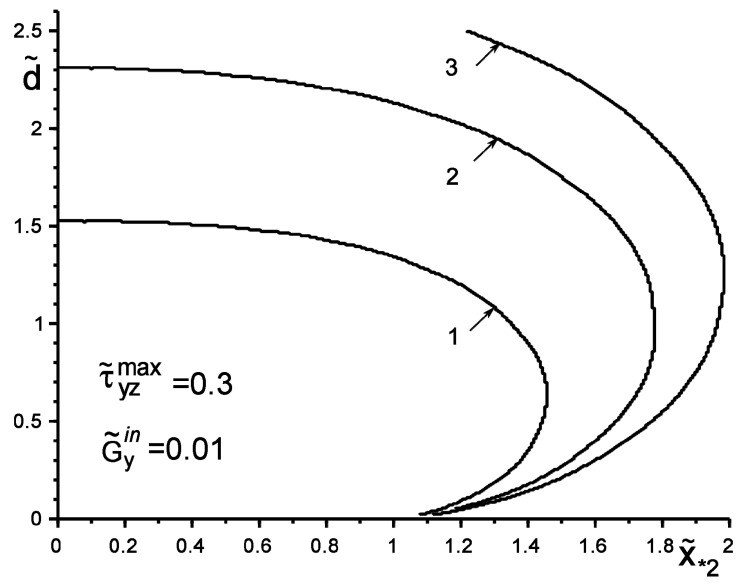
The boundary of the “safety zone” when loading the structure with a softer than the matrix inclusion by a concentrated force: 1—${\tilde{Q}}^{*}=0.5$; 2—${\tilde{Q}}^{*}=0.75$; 3—${\tilde{Q}}^{*}=0.9$.

**Figure 3 materials-14-04928-f003:**
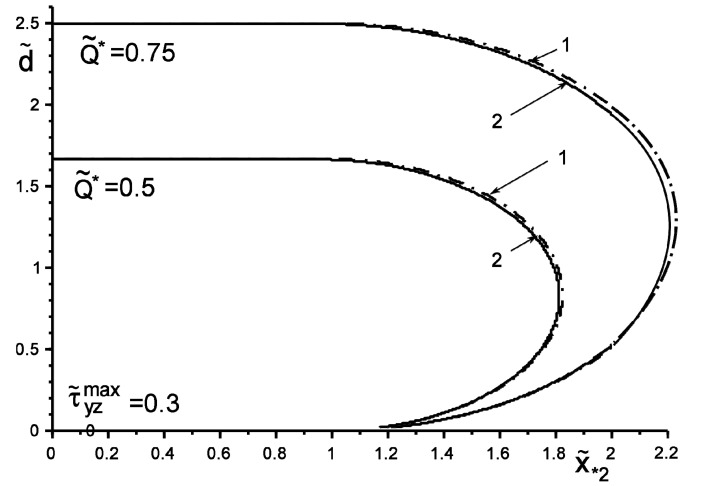
The boundary of the “safety zone” when loading with the concentrated force ${\tilde{Q}}^{*}$
of the structure without (1—${\tilde{G}}_{y}^{in}=1$) and with harder than matrix inclusion (2—${\tilde{G}}_{y}^{in}=10$).

**Figure 4 materials-14-04928-f004:**
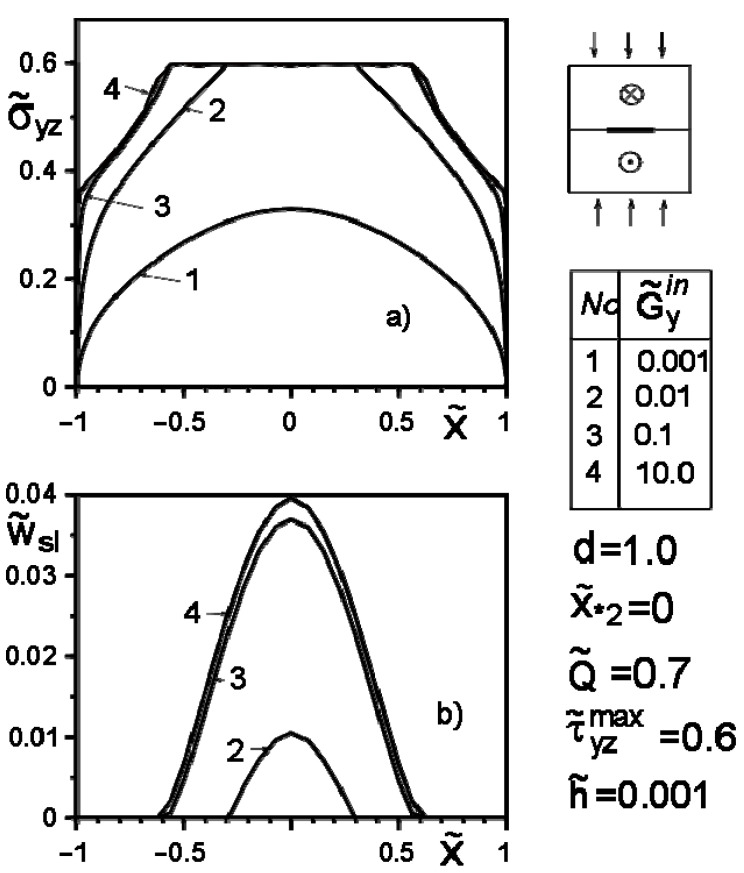
Stress distribution along with the inclusion–matrix boundary, (**a**) and the size of the slip zone (**b**) depending on the ratio ${\tilde{G}}_{y}^{in}/{\tilde{G}}_{k}$

**Figure 5 materials-14-04928-f005:**
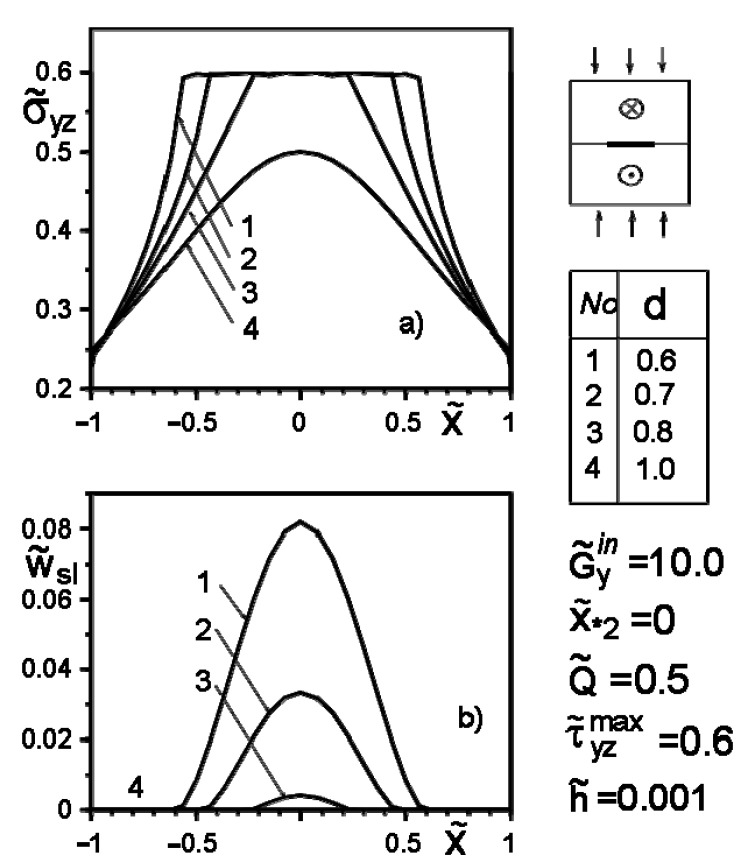
Stress distribution along with the inclusion–matrix boundary (**a**) and the value of the slip zone (**b**) for inclusion harder than matrix, depending on the distance of the force application point from its axis.

**Figure 6 materials-14-04928-f006:**
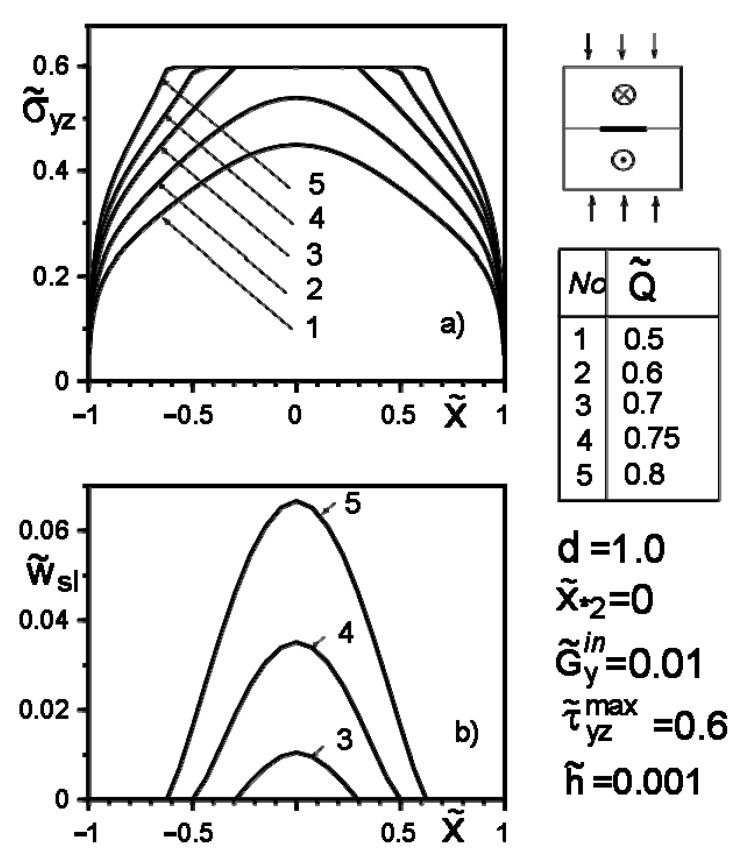
Stress distribution along with the inclusion–matrix boundary (**a**) and the size of the slip zone (**b**) for a softer than matrix inclusion as a function of force intensity growth.

**Figure 7 materials-14-04928-f007:**
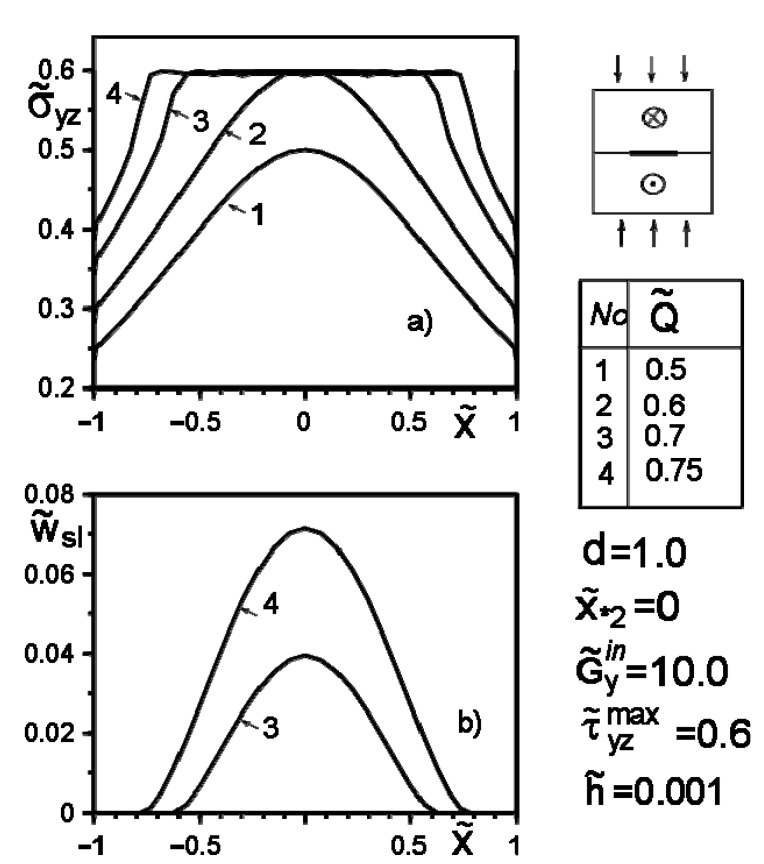
Stress distribution along with the inclusion–matrix boundary (**a**) and the value of the slip zone (**b**) for a harder than matrix inclusion as a function of force intensity growth.

**Figure 8 materials-14-04928-f008:**
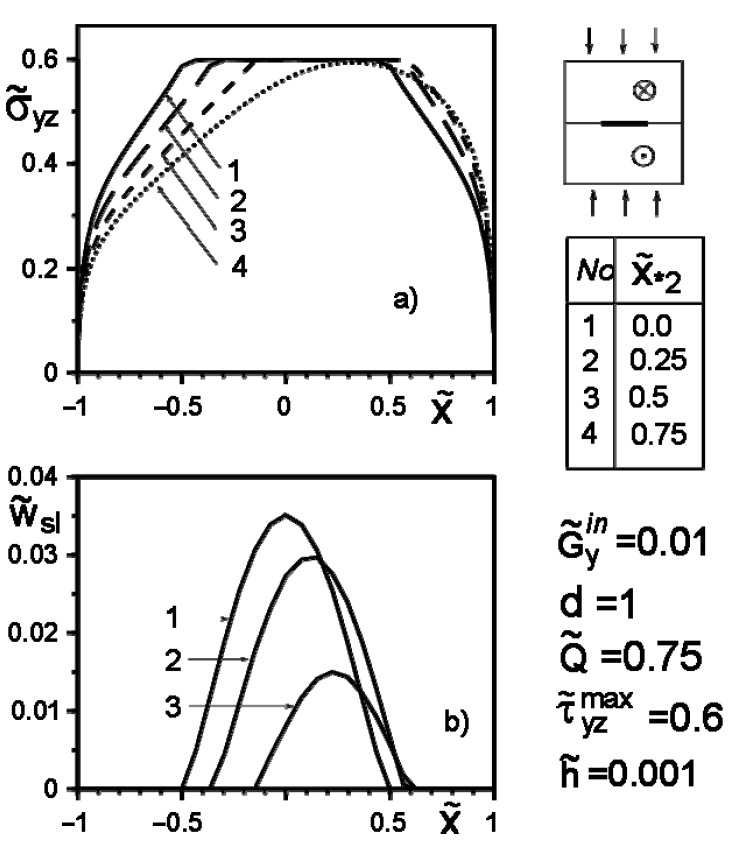
Stress distribution along with the inclusion–matrix boundary (**a**) and the value of the slip zone (**b**) for a softer inclusion than matrix depending on the change in the coordinates of the force application points along the inclusion axis.

**Figure 9 materials-14-04928-f009:**
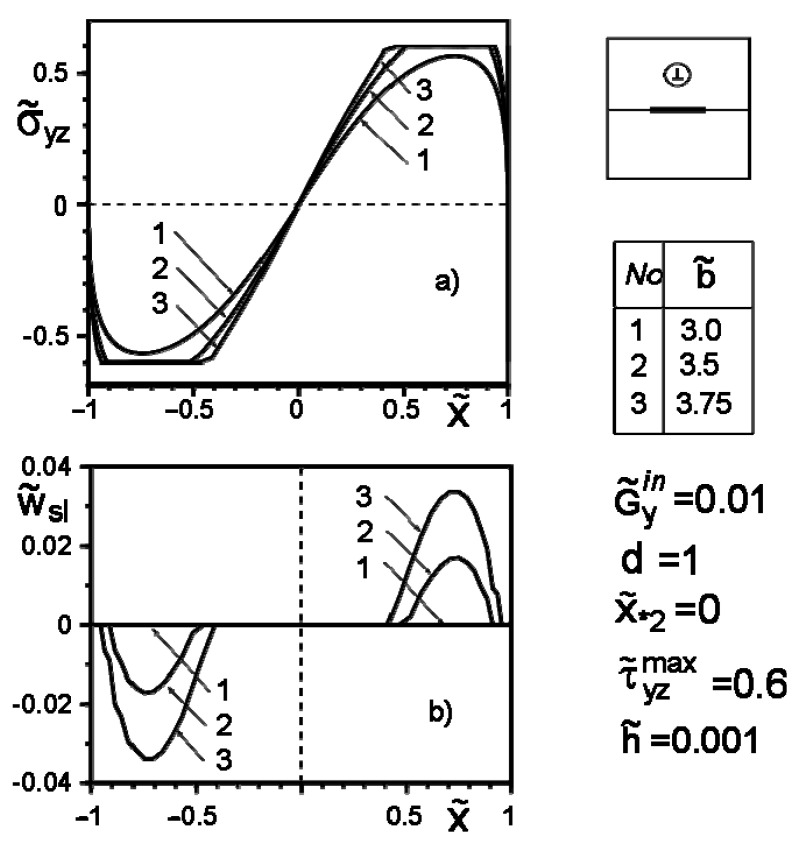
Stress distribution along with the inclusion–matrix boundary (**a**) and the value of the slip zone (**b**) for a softer than matrix inclusion under screw dislocation loading.

## Data Availability

The data presented in this study are openly available at DOI 10.1007/s11003-018-0114-2, reference number [6].

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
