# Peer review of "Effect of Frictional Slipping on the Strength of Ribbon-Reinforced Composite"

_materials, 2021, doi:10.3390/ma14174928_

Round 1

Reviewer 1 Report

Introduction provides sufficient background and include all relevant references.

Research design is appropriate. Methods are adequately described.

Results are presented and supported with Figures from numerical results.

Conclusions is supported by the results.

Spaces between text and equations are missing.          

Author Response

The paper focuses on the effect of frictional slipping on the strength of ribbon reinforced composite. The authors proposed a numerical-analytical method to determine the stress-strain state of bimaterial structures with interphase ribbon-like deformable inhomogeneities under combined force and dislocation loading. The derivation process is rigorous, and the obtained conclusions are convincing. Several important conclusions are obtained, e.g., with the friction coefficient, type, and place of load application unchanged, the slippage will start at a lower load and the slippage zone will be larger in the case of a more rigid from the matrix inclusion.

The proposed method and conclusions are expected to be used in the studies of the interaction of band inhomogeneties in composite structures. I think this paper can be published in journal of Materials. There are some minor problems needed to be addressed.

  1. The structurally modular method of jump functions has been constructed for biomaterial structures. It would be better if the method is verified in a given typical material system, e.g., graphene-loaded organic composite.

The SMJFM has been constructed for the problems of micromechanics of bi-material structures and calculations are made in the non-dimensional version, which allows us to extend the obtained analysis to any combination of the properties of the constituent materials. The variant of the graphene-loaded organic composite proposed by the reviewer was not considered by us, but we are grateful for the idea and in the future we will make a corresponding study.

  1. In Figure 1, in order to easily understand the model, more explanation can be given.

We agree, an appropriate addition to the explanation of the Figure 1 is included in the text of the article.

  1. The reference order in the maintext should be corrected.

Reference order corrected

We are grateful for an objective review

Our best regards, Heorhiy Sulym and Yosyf Piskozub

Reviewer 2 Report

The paper focuses on the effect of frictional slipping on the strength of ribbon reinforced composite. The proposed method in the paper is rigorous, which can be used to study in many materials such as the interaction of band inhomogeneties in composite structures. I think this paper can be published in journal of Materials. There are some minor problems needed to be addressed.

  1. The structurally modular method of jump functions has been constructed for biomaterial structures. In my opinion, the method needs to be verified in the given typical material systems.
  2. In Figure 1, in order to easily understand the model, more explanation can be given.

Author Response

(The authors gave the same response as above.)

Reviewer 3 Report

The problem of friction is often underestimated in micromechanics. The paper presents good analytical solutions to the problem depicted in Fig. 1.  It gives a very good reference for numerical solutions. In my opinion, the paper can be published as-is.

Author Response

We are grateful for an objective review

Our best regards, Heorhiy Sulym and Yosyf Piskozub

Reviewer 4 Report

Journal:  Materials MDPI

Ref manuscript ID   materials-1303766

Manuscript titleEffect of frictional slipping on strength of the ribbon reinforced composite”

Comments to authors:

In this paper, the authors developed a numerical-analytical approach to study structures with ribbon-like elements with possible frictional contact between the constituent elements.

the purpose of this approach is to determine the stress-strain state of bi-material structures with interphase ribbon-like deformable inhomogeneities under the combined force and dislocation loading

Important results were derived from thisanalysis .This is an excellent report dealing with significant technical matters. I find no fault what so ever with the methods, data analysis, or conclusions. The work, as with all work coming from this particular group, is fundamentally sound. My comments here are concerned solely with the organization of the manuscript.

I therefore think the manuscript needs improving in two ways:

  1. A) Restructuring and clarification for a general scientific audience.
  2. B) Additional methodological detail.

Consideration of these points will, I believe, lead to an improved report   illustrates the key concepts and conclusions

In my opinion, the manuscript is suitable for publication in Materials MDPI, after the authors have addressed the following comments and questions:

  • There are numerous places in the text with English grammatical errors. The authors should be full checking for grammar and mistakes to meet the quality of Journal. Indeed, there are many errors in this document, some words are missing in several places, punctuation was not observed along the paper also lacks a comma (,) and point in several places, some sentences have meaning disorders and should be checked.

 Please, there are many other errors in this document in several places, please carefully refine the English language

  • The use of definite and indefinite articles in the text requires extensive correction and revision ; please ,,use of articles 'The', ,’ the’ 'A' and 'An' needs to be corrected at some places
  • Please focus the abstract on your study and your results. In particular the last two sentence are vague. More generally, I suggest to focus the manuscript on the scientific results rather than on the innovation in engineering
  • In the document, several questions that occupy me with regard to the subject addressed by them and which I ask the authors to clarify in detail as answers:

What is reinforced composite material?

What are the advantages and disadvantages of ribbon reinforced composite?

What are reinforcement materials found in composites?

Which reinforcement is best for composites?

What is the reinforcement phase of a composite?

What are the 4 types of composites?

What is matrix reinforcement?

What is the function of reinforcement in composites?

  • At the end of the introduction section, the authors said this: “Separately, it should be noted that the vast majority of the works on the study of the structural stability of bodies with thin inhomogeneities does not cover the entire possible range of force or dislocation loading of structures, does not give a full opportunity to determine the critical load and the so-called "safety zone" for its application in order to optimize the properties of structures for certain types of loading.” , I strongly recommend that they clarify this statment in detail and in-depth discussion there.
  • The authors presented the results, but they didn't give a sufficient discussion for the results. This makes the paper look like a lab report rather than a research paper.
  • From the introduction, it seems that the authors know little about what has been done by others in this field. A detailed literature review is strongly recommended. 
  • There are many research papers study the same problem which investigated in the present paper. What is exactly the new point of this work?
    The authors should focus to clarify this issue in the paper.
  • Authors should explain more about the novelty of their work which is not clear in introduction,the work objective it is not clearly written.
  • A simple flow chart is required to represent the steps-wise procedure followed for carrying out the analytical analysis.
  • Key assumptions and their implications could have been elaborated
  • The quality and form of all equations inserted in the document should be greatly improved in a new edition
  • The authors should think over the real significance of their results and try to rewrite this section to improve understanding of the conclusions
  • The quality of the figures in this document needs to be improved; the figures need to be larger in size so the data and labels can be clearly read.
  • The introduction section is adequate but needs some language revision.
  • The literature list should be expanded by adding extra work related to the subject.

In results and discussion, the authors should discuss on their results deeply. I strongly recommend expanding: Introduction, Conclusions and the Results sections. The aim should be to: 1) give a broader view of the literature on the topic and the current state-of-the-art; 2) clarify and discuss the novelty and the significance of the results obtained here, and compare them with those available in the literature, also including discussions on potential applications; 3) complete the manuscript with some additional, less basic results; 4) The authors should show the comparison between their results and previous works.
The following are the valuable studies to make the introduction section more concise to show the previous literature:

-          An approximate numerical solution to the Graetz problem with constant wall temperature, International Journal of Computing Science and Mathematics, 2017 Vol.8 No.1, pp.35 - 51

  • A thermomechanical model for the analysis of disc brake using the finite element method in frictional contact, Journal of Thermal Stresses, 43(3) 305-320 (2020)
  • . In 'Result and Discussion' authors have noted observations. But it is suggested that to provide physical explanations of all obtained results which can enrich the quality of the paper.
  • The authors must provide a greater discussion of the results
  • The conclusion should be clear.
  • The authors must explain to us whether they had used the equations inserted in the simulation procedure adopted in their work or as only bibliographic references
  • Validation of present solution with the others already existing study must be added.

Altogether, the paper needs modification to be suitable for the standards required for publication; therefore I recommend that it required to “Major revision”.

I look forward to receiving the revised version of this manuscript

My best regards

Author Response

We are grateful for the very thorough review and recommendation of the reviewer. It is a very valuable insight, which we will certainly take into account.

Our best regards, Heorhiy Sulym and Yosyf Piskozub

Round 2

Reviewer 4 Report

Journal:  Materials-MDPI

Ref manuscript ID  Materials-1303766

Manuscript title “

 Effect of frictional slipping on strength of the ribbon reinforced composite”

Comments:The authors have satisfactorily responded to all my questions and made the necessary changes to the manuscript. The revised version of the manuscript appears to be good.

My final recommendation: Accept With No Changes

Thank you again for giving me a chance and inviting me to review this document.

Best regards